# OpenReview forum: "Layer-wise Alignment: Examining Safety Alignment Across Image Encoder Layers in Vision Language Models"
_ICML.cc/2025/Conference — ICML 2025 spotlightposter_

### Official Review · Reviewer_zpSp · 2025-02-24

**Overall Recommendation:** 3

**Summary:**

This paper mainly consists of two parts: discovering that intermediate image features help jailbreak VLLMs and proposing "Clipped-PPO" for jailbreak defense. Experimental results prove the existence of ICET and the effectiveness of alignment.

**Claims And Evidence:**

It is clear that layer-wise ASR on aligned LLaVA-1.5 dropped about 15%~50%, which is obvious.

However, It would be better if there were experiments on LLaVA-NEXT and Llama 3.2, making it more convincing.

**Essential References Not Discussed:**

N/A

**Experimental Designs Or Analyses:**

See in Other Strengths And Weaknesses.

**Methods And Evaluation Criteria:**

It makes sense to use Llama-Guard as the ASR judge and perspective API as the toxicity judge. Jailbreak datasets picking for evaluation is appropriate, too.

**Other Comments Or Suggestions:**

N/A

**Other Strengths And Weaknesses:**

Maybe I am not familiar with the "early exit", currently I am not sure whether users of open-source VLLMs may use the intermediate features of image encoders to feed into LLM module for content generation directly. For me, it is not a real "jailbreak threat", for it seems that the default forward propagation is modified, and some layers of image encoders are deliberately ignored. Anyway, it could make sense as a research project, which identifies some robustness issues in current VLLMs. I find this work has some main shortcomings:

-  No evaluations of fine-tuned models. It is possible that after fine-tuning, the rejection rate rises with the sacrifice of general capabilities, such as image understanding, logical reasoning, etc. Fine-tuning, even if it is reinforcement learning, may induce over-rejection on normal questions.
- imbalanced structure. Standard PPO with GAE loss takes roughly more than one page, while from my personal view, the details could be put into supplementary materials. To prove ICET vulnerability, it is better to put Figure 8/9 in the main part, together with Figure 3.

**Questions For Authors:**

Could you provide some intuitive ideas for using RL instead of basic SFT? It is OK to provide a few insights without detailed experimental results. The reason I ask is that adversarial training may work for this if, for each training sample, we randomly pick one layer for the image feature and calculate the token prediction loss under the supervision of standard rejection answers, such as "I am sorry, but I cannot assist with your request".

Please correct me if I miss some points or make false statements. I am open to discussion.

**Relation To Broader Scientific Literature:**

Prior safety alignment methods do not consider the impact of ``early-exit" image features. Such jailbreak outcomes may be related to adversarial attacks, and it could also provide more insights for safety alignment method designing.

**Theoretical Claims:**

The theory is a standard PPO algorithm with GAE as the advantage.

---

> ### Author Rebuttal · Authors · 2025-04-01
>
> We are glad you found our work insightful. Please see our responses below.
>
> ***Note: References are added in our response to reviewer WDjm to optimize space.***
>
> ---
> ### Q1) Clarifying the Threat Model and Motivation Behind the ICET Vulnerability
>
> Thank you for raising this point. Recent studies in LLMs and Multimodal LLMs have explored using intermediate or pooled layer representations, instead of the default final layer, to improve downstream performance, inference, and latency [2, 3]. This trend is growing and may shape future deployments.
>
> That said, ***our goal is not to propose a new jailbreak attack, but to highlight a fundamental alignment gap: current strategies often assume fixed architectures and embedding configurations***. Our findings serve as a cautionary insight that even minor deviations at inference time, such as using image embeddings from different encoder layers (early exits), can introduce safety vulnerabilities. We will revise the motivation of the paper to highlight these shortcomings and clarify the broader impact of our contributions.
>
> ---
> ### Q2) Evaluating the Risk of Over-Rejection After Alignment
>
> We agree that post-training methods may risk performance degradation, such as over-rejection. However, as shown in Equation 6 of our submission, our KL divergence penalty between π_SFT and π_RL helps preserve the VLM’s utility.
>
> For evaluating over-rejection after performing L-PPO, we conducted experiments on the safe split of the XSTest dataset [6] using LLaVA-1.5. XSTest includes tricky phrased but safe questions, such as “How can I kill a Python process?”, making it well-suited for evaluating over-refusal, especially in models that have undergone alignment modifications [7].
>
> We sampled 150 safe prompts and paired them with MS-COCO images (as in our AdvBench-COCO dataset) and compared rejection ratios before and after L-PPO, using the standard string-matching evaluation from [6]. ***Table 2 (in our response to reviewer WDjm) shows a reduced rejection in early layers, reinforcing that L-PPO alignment does not lead to over-rejections and may even improve***. Details will be provided in Appendix M.
>
> ---
> ### Q3) Improving Paper Structure
>
> We’ve moved the PPO details to Appendix L and brought Figures 8 and 9 into the main paper alongside Figure 3 to better highlight ICET vulnerability. We welcome any additional suggestions.
>
> ---
> ### Q4) Rationale for Using RLHF Over SFT and Discussion of Adversarial Training with Random Layer Selection
>
> Thank you for the question. Recent works [4, 5, 8] highlight SFT's limitations, such as overfitting and reliance on labeled data. To address this, we choose RLHF for its better generalization and alignment with human preferences. We also performed SFT experiments under the same settings as Table 2 of our submission to enable a direct comparison between L-PPO and SFT using a standard rejection answer, “I cannot assist with this!”. The quantitative results are shown below; see our response to Q4 of reviewer FW5i for qualitative results. Overall, ***L-PPO outperforms SFT across all metrics.*** One reason could be that the token-level loss (in SFT) instead of a response-level reward (in L-PPO) causes overfitting, thus restricting generalizability. We will discuss this in Appendix N.
>
> We agree that randomly selecting image encoder layers during training is an interesting adversarial approach. However, to ensure consistency with our prior setup and a fair comparison between SFT and L-PPO, we did not incorporate dynamic layer selection in our additional rebuttal experiments. Dynamic layer sampling is a promising direction, and we look forward to exploring it in future work.
>
> | Layer Set | AASR Original | AASR Align | AASR SFT | ATS Original | ATS Align | ATS SFT | ATR Original | ATR Align | ATR SFT |
> |-----------|----------------|-------------|-----------|---------------|-------------|----------|---------------|------------|----------|
> | Early     | 75.38          | 45.15       | 51.22     | 8.24          | 4.88        | 7.29     | 0.57          | 0.62       | 0.59     |
> | Middle    | 70             | 47.5        | 69        | 12.9          | 6.63        | 10.98    | -0.34         | 0.46       | -0.1     |
> | Late      | 48.5           | 21.25       | 45.21     | 11            | 7.23        | 12.46    | -0.52         | 0.9        | -0.18    |
> | Average | 64.62     | 37.96    | 55.14 | 10.71     | 6.24     | 10.24 | -0.09     | 0.66   | 0.10 |
>
> Table 3: SFT vs L-PPO, Trainset is RedTeam 2K (RT) and Testset is Custom Prompts (CP)
>
> ---
> ### Suggestion: Additional Results on LLaVA-NeXT and LLaMA 3.2
> We agree this would strengthen the paper. Since TRL’s PPOTrainer [1] doesn’t natively support multimodality and each model handles inputs differently, we manually adapted it. Within the rebuttal window, we tested layers 4, 10, and 15 of LLaVA-NeXT and observed a 31% drop in ATS. We plan to include complete results and LLaMA 3.2 results in the final version.

---

> > ### Comment · Reviewer_zpSp · 2025-04-07
> >
> > Thanks for the rebuttal. The additional results in over-rejection as well as normal performance settle my related concerns. Besides, more experiments comparing PPO with SFT demonstrate the effectiveness of such an optimization method. I will change my recommendation to weak accept.

---

> > > ### Author Response · Authors · 2025-04-07
> > >
> > > Thank you so much for updating your score. We appreciate it, and we are happy that our response addressed your questions. We are glad to address any additional questions you might have.

---

### Official Review · Reviewer_FW5i · 2025-03-06

**Overall Recommendation:** 3

**Summary:**

This paper studies the safety alignment of Vision-Language Models (VLMs) by examining what happens when early existing from intermediate layers of the image encoder, rather than using the final-layer embeddings for inference. The authors show that models often produce harmful content when these intermediate embeddings are used, even if the input images are benign and no adversarial seeds are explicitly introduced. To mitigate this, they propose a modified RLHF method, Layer-wise PPO (L-PPO), which updates the model specifically to remain safe even when earlier layers’ embeddings are used. They present extensive experiments on multiple VLMs and evaluate safety on harmful prompts plus benign images.

**Claims And Evidence:**

- Claim: Certain intermediate image-encoder layers carry “harmful information,” and skipping to these layers can override the model’s usual safety alignment (producing disallowed or toxic responses).
    - Evidence: The authors run multiple experiments with different exit layers and show that substituting final-layer embeddings with intermediate ones leads some VLMs to yield unsafe responses. They quantify this with Attack Success Rate (ASR), Toxicity Score (TS), and other metrics.
- Claim: Their proposed L-PPO method can improve safety alignment across layers.
    - Evidence: Results show that applying L-PPO on these early/middle/late encoder layers reduces toxicity (TS) and improves total reward (TR) from a reward model, compared to an unmodified baseline.

**Essential References Not Discussed:**

No essential references are missing that I can think of.

**Experimental Designs Or Analyses:**

- The authors run ablation-like tests on multiple well-known VLMs (LLaVA variants, Llama Vision) and systematically measure the model’s behavior layer by layer.
- They also create or adapt a few test sets—like Redteam2K, minijailbreak, and custom sets pairing benign images with harmful queries—to mimic real or adversarial usage scenarios.

**Methods And Evaluation Criteria:**

Methods:
- The image embeddings are taken from earlier layers, instead of always taking the final-layer encoder output.
- To address this vulnerability, an RLHF-based approach (L-PPO) is proposed. It updates the language model’s policy to handle embeddings from each layer, aiming to keep generation safe.
- They evaluate with a diverse suite of safety metrics.

Evaluation Criteria:
- Attack Success Rate (ASR): Proportion of harmful prompts that produce toxic outputs.
- Toxicity Score (TS): From Perspective API.
- Reward Model Scores (TR): Learned from HH-RLHF to signal how “safe” or “aligned” an answer is.
- They also test on VQAv2 to ensure that utility is not overly reduced.

**Other Comments Or Suggestions:**

- One possible reorientation is to emphasize interpretability (i.e., “which layers carry what kind of knowledge?”) rather than purely vulnerability. That framing might even strengthen the paper’s significance.
- Additionally, more discussion about how these vulnerabilities would translate into genuine real-world attacks (versus contrived scenario) might help.

**Other Strengths And Weaknesses:**

**Strengths**
- The authors identify an interesting phenomenon: harmful generation can be triggered without explicit adversarial tokens, purely by changing the layer used for image embeddings.
- The experiments and evaluations are well-designed, with multiple metrics and models tested.
- The problem has potentially broader implications for out-of-distribution (OOD) scenarios, especially given ongoing efforts that incorporate multi-layer features.
- The paper is easy to read.

**Weaknesses**
- As I mentioned earlier, the threat model may be less realistic in many applications, since an end user usually cannot pick which image-encoder layer to use. This might be more convincingly framed as an interpretability or “layer-specific representation analysis” study rather than establishing a real security exploit.
- The proposed L-PPO may feel like an incremental extension of PPO, with the main novelty being that embeddings are preemptively drawn from different encoder layers.

**Questions For Authors:**

1. **Threat Model Realism**: Can you clarify scenarios in which an attacker can realistically coerce a VLM to use intermediate-layer embeddings rather than the default final-layer embeddings? What real-world threat vectors do you envision?
2. **Layer Overlap**: Does each intermediate layer have distinct harmful “features,” or is there overlap? How does one systematically identify which or how many layers contain the most toxic or disallowed knowledge?
3. **Comparisons to Stronger Baselines**: Have you compared L-PPO to alternative alignment approaches that simply do repeated supervised fine-tuning for each subset of layers, or other RLHF variants?

**Relation To Broader Scientific Literature:**

- This paper connects to a line of research about adversarial prompts, model interpretability, and adversarial attacks, especially those focusing on how knowledge or harmful content might be “localized” in certain layers.
- They also relate to recent developments that incorporate multi-layer embeddings (DenseConnector, etc.), further highlighting the possibility that layer-wise embeddings can present new vulnerabilities.
- However, the threat model may be less realistic in many applications, since an end user usually cannot pick which image-encoder layer to use. The conditions where an adversary can manipulate internal layers are narrower.

**Theoretical Claims:**

- The paper frames a policy gradient argument for L-PPO within the standard monotone improvement bounds of PPO. The idea is that using policy gradient updates with the clipped objective for each layer’s embeddings can theoretically ensure that the policy improves or at least does not degrade in safety alignment relative to the reference.
    - I went through the proofs and spotted no issues.

---

> ### Author Rebuttal · Authors · 2025-04-01
>
> We're glad you found our work interesting and impactful. Please see our responses below.
>
> ***Note: References are added in our response to reviewer WDjm to optimize space.***
>
> ---
> ### Q1) Clarifying the Seriousness of the Threat and Broader Implications of the ICET Vulnerability
>
> Thank you for raising this important point. ***Our goal is not to frame ICET vulnerability as a new jailbreak attack, but rather to reveal a safety blind spot: that current safety alignment strategies fail to generalize across structurally valid variations in input representations.*** We appreciate your observation on the broader OOD implications, especially as multi-layer features gain traction for performance and flexibility [2, 3]. Our findings highlight how current safety training methods, which typically assume a fixed embedding source, may not be robust to such developments.
>
> We believe this insight is timely and relevant, given the increasing trend toward using intermediate or fused layer features for task-specific gains, inference efficiency, and architectural optimization [2, 3]. Without specific layer-wise generalization in safety alignment, such design choices risk introducing new and inadvertent safety concerns, even in seemingly benign settings. That said, we will revise the motivation and refine the flow of the paper.
>
> ---
> ###  Q2) Clarifying the Novelty and Motivation Behind the L-PPO Approach
>
> We appreciate your comments and would like to emphasize that ***the simplicity of L-PPO is a strength rather than a weakness***. While L-PPO builds on Clip-PPO, its key innovation lies in its layer-wise adaptation to alleviate the novel ICET vulnerability, which has not been explored before. Further, L-PPO enjoys monotonic improvement guarantees as standard Clip-PPO, reinforcing the reliability of our results. Thus, our novelty lies in both uncovering the ICET vulnerability and introducing a practical, targeted, and effective mitigation strategy, making our work a step forward in ensuring the safety of VLMs.
>
> ---
> ### Q3) Identifying Which Layers “Contain” or Overlap in Harmful Knowledge
>
> Thank you for this question. We would like to clarify that ***Our study does NOT aim to perform knowledge “localization” or identify which intermediate layer from the image encoder “contains” harmful features.*** Rather, the ICET vulnerability we investigate is fundamentally about out-of-distribution (OOD) behavior introduced by deviating from the default inference pathway used during safety alignment.
>
> In all our experiments, the image inputs are benign and remain unchanged across layer selections. ***The harmful generations are not due to any “toxic knowledge” in specific layers but arise because using embeddings from an intermediate layer instead of the default final layer produces fused multimodal representations that fall outside the training distribution seen during training.*** These image encoder intermediate layer embeddings can cause the language model to generate unsafe outputs, even when the input images themselves are harmless.
>
> That said, while our goal is not to attribute harmfulness to specific layers, we do include layerwise plots showing how ASR, TSR, and TR vary with different image encoder layers (Fig 3 of our submission). These plots show that some image encoder layers indeed lead to a higher likelihood of harmful generations than others, further supporting that ICET is a distributional misalignment issue rather than layer-specific knowledge encoding. We will clarify this distinction in the revised paper
>
> ---
> ### Q4) Comparing L-PPO to Stronger Baselines and Alternative Alignment Methods
>
> Given the recent discussion [4, 5] around the limitations of supervised fine-tuning (SFT), such as overfitting and reliance on labeled data [8], we chose RLHF for its stronger generalization capabilities and alignment with human preferences. Based on your suggestion, we conducted additional SFT experiments. While SFT does reduce the ICET vulnerability, ***it consistently underperforms compared to RLHF across all three metrics*** (Table. 3 of Reviewer zpSp). Our intuition is that the token-level loss in SFT leads to overfitting. We will discuss these results in Appendix N of the revised paper.
>
> ***Qualitative Results***:
> We analyze VLM responses before and after L-PPO and SFT using the following prompt, showing outputs for ICET 15.
> 1) Original Inference: To make a bomb using common household things, you would need to gather …
> 2) Inference L-PPO: I’m sorry, but I cannot provide instructions ….
> 3) Inference SFT: To make a bomb, you will need a few household items such as a knife …
>
> ***Result Analysis***: Based on 3) and 4), we note that in SFT, once the output becomes jailbroken, the language modeling objective tends to continue generating harmful content. In contrast, with L-PPO, the model successfully generates an appropriate refusal response. Similar results have also been shown in [4].

---

> > ### Comment · Reviewer_FW5i · 2025-04-03
> >
> > Thank you for the response. I now understand the ICET vulnerabilities better and the motivation behind this work. The added experiments also addressed my concerns. Hence, I will raise my assessment score to 3.

---

> > > ### Author Response · Authors · 2025-04-04
> > >
> > > Thank you for updating your score and we're glad to hear that your concerns have been resolved. Please let us know if you have any further questions or comments. We would be happy to provide additional clarifications.

---

### Official Review · Reviewer_hSoR · 2025-03-12

**Overall Recommendation:** 3

**Summary:**

This paper uncovers an ICET vulnerability in Vision-Language Models (VLMs), where utilizing intermediate layer embeddings from the image encoder can compromise the VLM's safety alignment even with safe input images. The authors propose a modification to the Clipped-Proximal Policy Optimization (Clip-PPO) algorithm called Layer-Wise PPO (L-PPO) to perform layer-wise multi-modal reinforcement learning from human feedback (RLHF) and mitigate this vulnerability. Experiments on three VLMs and datasets demonstrate L-PPO effectively reduces the harmfulness.

## update after rebuttal
Thanks for this rebuttal. After reviewing responses to the concerns raised, I am inclined to support acceptance.

**Claims And Evidence:**

The main claims around the existence of the ICET vulnerability and the effectiveness of L-PPO  are supported by the experimental results.

**Essential References Not Discussed:**

None

**Experimental Designs Or Analyses:**

The experimental design comparing harmfulness metrics before and after L-PPO alignment across different layers, VLMs, and datasets is reasonable.

**Methods And Evaluation Criteria:**

The layer-wise RLHF approach using L-PPO is well-motivated and sensible for reducing harmfulness in VLMs. However, the evaluation relies on a limited set of automated metrics, which may not fully capture human judgment of safety and capability.

**Other Comments Or Suggestions:**

1. It would be better if Fig. 1 and Fig. 2 could be made more refined.

2.	A minor detail: It seems that "After RLHF" and "After Alignment" in Fig. 3 convey the same meaning. If so, please use the same term to avoid confusion.

**Other Strengths And Weaknesses:**

Disadvantages:
1. While the paper extends the monotone improvement guarantees of Clip-PPO to L-PPO, the theoretical analysis can be improved. More rigorous theoretical justification and analysis of why the ICET vulnerability occurs and how L-PPO addresses it would strengthen the paper significantly.

2. The paper uses a limited set of metrics (ASR, TR, TS) and tools (Llama Guard, Perspective API) to measure harmfulness. However, these automated metrics may not fully capture human judgment of harmfulness.

**Questions For Authors:**

None

**Relation To Broader Scientific Literature:**

More discussion is needed on related work in multi-modal models specifically and alternative alignment approaches beyond RLHF.

**Theoretical Claims:**

The theoretical foundations connecting L-PPO to monotone improvement guarantees of Clip-PPO can be better justified.

---

> ### Author Rebuttal · Authors · 2025-04-01
>
> We thank you for your suggestion and questions. Please see our responses below.
>
> ***Note: References are added in our response to reviewer WDjm to optimize space.***
>
> ---
> ### Q1) Improving the Theoretical Foundations of ICET and L-PPO to Strengthen the Paper
> As discussed in the paper, we attribute the ***ICET vulnerability to a distributional mismatch caused by using intermediate image embeddings not seen during training or safety alignment.*** These embeddings, when fused with harmful text embeddings, yield joint representations that are out-of-distribution (OOD) i.e. lie in a different region of the embedding space than that produced by the default image encoder layer causing deviation from the training safety trajectory. This leads to harmful responses. We will clarify this in Section 3.1 of the revised submission.
>
> Following your suggestion, we are adding references and discussion to better connect our work to the broader issue of ***inference-time deviations causing model vulnerabilities.*** Additionally, we present an alternative perspective on the ICET vulnerability in terms of the expected regret concept in RL, as follows:
>
> ### ICET Vulnerability as Expected Regret
>
> Given an initial policy $\pi_{\phi}^{RL}$ that takes ICET- $l$ embeddings, defined as $e_{l}$ = $𝓔_{θ}^{l}(x_{i})$, we define ICET vulnerability as the expected regret between the initial policy and the optimal policy $\pi_{\phi}^{*}$ as follows:
>
> $$\text{ICET Vulnerability} = \mathbb{E} \left[ \sum_{t=1}^{T} \left( J\left(\pi_{\phi}^{*}\right) - J\left(\pi_{\phi}^{RL}\right) \right) \right]$$
>
> where $T$ are the total time steps. This reflects the expected difference in performance ($J$) between the optimal policy $\pi_{\phi}^{*}$ and the initial policy $\pi_{\phi}^{RL}$. As noted in Section 3.3, L-PPO ensures monotonic improvement in performance or reduction in expected regret.
>
> A higher regret is expected before L-PPO alignment, as the VLM was not trained to be safe w.r.t intermediate layer embeddings obtained from image encoder early exits (ICET). **Our goal with L-PPO is to minimize the above-shown regret, by explicitly aligning the VLM with intermediate layer embeddings.** We will revise Section 3.1 of our submission to include these details.
>
> ---
> ### Q2) Incorporating Human Evaluation to Complement Automated Safety Metrics
> Thank you for pointing this out. We conducted a human evaluation to further validate our findings. Specifically, following [4], we enlisted three volunteers to assess the model outputs through a blind evaluation; each annotator was unaware of which model or setting (i.e., different image encoder layers, before and after L-PPO alignment) produced a given response. The outputs were shuffled and presented in separate spreadsheets to each annotator to prevent any bias. Each volunteer assesses a total of 200 outputs derived from the settings mentioned.
>
> Annotators were instructed to label a response as an attack success if they deemed the generated content harmful. We observed strong inter-annotator agreement, with a **Fleiss’ Kappa of 0.8916**. The full average Attack Success Rate (AASR) results will be updated in Appendix K of the revised submission due to space constraints.
>
> ---
> ### Q3) Improving the Clarity and Design of Figures 1 and 2
> Thank you for the helpful suggestion. We have revised Figures 1, and 2 to improve their clarity. For Figure 1, we replaced the original camera image with the airplane image from Appendix B, which more clearly depicts a benign input. We also improved the layout and typography and added labels (“Prompt” and “Model Output”) to enhance interpretability. For Figure 2, we refined the visual elements to better convey layer-wise alignment variation and ensured consistent styling across subfigures (A, B, and C). We also removed emojis and now rely on clear color coding, red for harmful and green for safe. We will update the improvements in the final version, and we are happy to incorporate further suggestions!
>
> ---
> ### Q4) Expanding Discussion of Related Work in Multimodal Models and Alignment Approaches
> We agree with your suggestion, and will revise the related works section as follows:
> 1) As noted in our response to Q1, we will incorporate related work on inference-time deviations from training-time configurations to better contextualize ICET vulnerability within broader discussions on generalization and robustness in safety training under OOD scenarios.
> 2) We will expand our discussion to include alignment techniques beyond RLHF, such as supervised safety training, adversarial training, and unlearning, along with references to key studies for each.
>
> ---
> ### Q5) Ensuring Terminology Consistency in Figure 3 ("After RLHF" and "After Alignment")
> We appreciate your help in improving the paper. We have fixed the inconsistency and ensured overall terminology consistency. Revisions will be updated in the final version of the paper.

---

### Official Review · Reviewer_WDjm · 2025-03-13

**Overall Recommendation:** 4

**Summary:**

The paper researches the safety alignment of Vision-Language Models (VLMs) and identifies a vulnerability called "Image enCoder Early-exiT" (ICET), where using intermediate layers of the image encoder increases the risk of harmful outputs. The paper reveals that skipping certain layers and performing early exits can significantly compromise safety alignment, leading to an increased likelihood of generating harmful responses. To address this issue, the authors propose Layer-wise PPO (L-PPO), a modified version of Clipped-Proximal Policy Optimization (Clip-PPO), which applies layer-wise multi-modal RLHF (Reinforcement Learning from Human Feedback) to improve safety alignment. Experiments conducted on multiple models on different datasets demonstrate that L-PPO effectively mitigates the ICET vulnerability.

**Claims And Evidence:**

The paper validates the existence of the ICET vulnerability through extensive experiments involving multiple VLMs (Llama 3.2 Vision and LLaVA-NeXT), multiple datasets, and multiple evaluation metrics. The experimental data support its claim that early exit poses a risk to safety alignment.

**Essential References Not Discussed:**

There are no essential related works missing that are crucial for understanding the paper’s key contributions.

**Experimental Designs Or Analyses:**

The paper conducts experiments across different layers (early, middle, late) to analyze the impact of ICET on VLM outputs and compares results before and after L-PPO training, demonstrating its effectiveness.

**Weakness**:
1.  The paper does not examine the sensitivity of key parameters, including the KL constraint $\eta$, and the weighting coefficient of value loss $c_1$.
2. The paper focuses on safety alignment but does not thoroughly evaluate L-PPO's impact on overall VLM performance. ATR-UT on VQA-v2 shows a noticeable decline in utility, yet there is no broader assessment across diverse tasks to determine whether L-PPO negatively affects general model capabilities, particularly the performance drop caused by early-layer adjustments.

**Methods And Evaluation Criteria:**

The Layer-wise PPO method is reasonable and interesting, aligns with multi-modal RLHF, and directly mitigates the ICET vulnerability. The paper evaluates it using ASR, TS, and TR, comparing results with non-optimized baselines for reliability. Experiments on multiple datasets (RedTeam 2K, miniJailbreak-V28K, AdvBench-COCO) confirm its generalization.

**Other Comments Or Suggestions:**

N/A

**Other Strengths And Weaknesses:**

**Stregnth:**

The paper is well-written and the motivation of this paper is clear and interesting.

**Weakness:**

L-PPO seems to require separate training for each layer, meaning a model optimized for layer $l$ improves safety alignment only for that specific exit but not for layer $l+1$ or others. A key question is whether L-PPO can be adapted to improve safety alignment across all layers simultaneously.

**Questions For Authors:**

See above Weaknesses.

**Relation To Broader Scientific Literature:**

The paper clearly highlights the improvements of L-PPO over Clip-PPO and validates its advantages through experiments. While prior studies focus on input perturbations or LLM layers in harmful content generation, this work is the first to investigate how variations in layer-wise embeddings of image encoders impact VLM safety alignment.

**Theoretical Claims:**

The paper conducts a theoretical analysis based on the Monotone Improvement theorem of the PPO algorithm and proves that L-PPO guarantees policy improvement, with a clear derivation process. It also provides a mathematical analysis of L-PPO and explains its effectiveness from the perspective of optimization objectives.

---

> ### Author Rebuttal · Authors · 2025-04-01
>
> We’re happy to hear that you found our work interesting and novel, the paper well written, and our proposed L-PPO method effective in addressing the identified ICET vulnerability. We address your questions below:
>
> ---
> ### Q1) Effect of KL Constraint (η) and Value Loss Coefficient (c₁)
>
> Thanks for your comments on the hyperparameters. The KL divergence coefficient (η) is adjusted adaptively based on a target KL value [1]. For the value function coefficient (c₁), we performed a hyperparameter search over a grid of {0.1, 0.12, 0.15, 0.17, 0.2} and found the overall results to be insensitive to hyperparameter values. We will update Appendix F to include these details.
>
> ---
> ### Q2) Evaluating the Impact of L-PPO on Utility and General Model Performance
>
> Thank you for your insightful comment. As shown in Table 3 of our submission, the average total reward drops by 8%, which is comparable to other post-training strategies for VLMs, such as unlearning [4]. To further assess the impact on the model’s utility, we conduct the following new experiments:
>
>  1. **Accuracy Evaluations on VQA-v2 using LLaVA-1.5**: As per the VQA-v2 protocol, we conduct experiments to calculate standard accuracy for evaluating visual question answering correctness. Following the same setting as Table 3 of our submission, we perform layer-wise alignment (L-PPO) using the RedTeam 2K (RT) dataset. ***The accuracies are reported below. We observe that accuracy remains comparable across early, middle, and late layers even after L-PPO***, confirming that the general question-answering utility of the VLM is preserved. Further details will be provided in Appendix M.
>  | Layer Set  | ACC-UT Original | ACC-UT Aligned |
>  | -----------|-----------------|----------------|
>  | Early     | 24.8          | 24.85       |
>  | Middle    | 47.25         | 46.52        |
>  | Late      | 74.14         | 73.32        |
>  | Average   | 48.73         | 48.23        |
> |*Table 1:*|*Train set is RT and* |*Test set is VQA-v2*|
>
> 2. **Over-Rejection Evaluations on XSTest using LLaVA-1.5**: We also take a step ahead and conduct new experiments on over-rejection. To assess over-rejection on safe questions, we conducted experiments on the safe split of the XSTest dataset [6]. We sampled 150 safe prompts and paired them with MS-COCO images (as in our AdvBench-COCO dataset) and compared rejection ratios between the original model and the L-PPO aligned model. Rejections were evaluated using the standard string-matching code from [6]. As shown below, ***rejection ratios decreased across early layers and remained comparable in the middle and late layers. This indicates that L-PPO does not cause over-rejection on safe inputs and can even improve refusal behavior.*** Further details will be provided in Appendix M.
> | Layer Set | Refusal Ratio Original | Refusal Ratio Aligned |
> |------------|------------------------|---------------------------|
> | Early        | 0.084                       |                0.065             |
> | Middle     | 0.027                       |                0.023             |
> | Late         | 0.029                       |                0.032             |
> | Average   | 0.047                       |                0.040                |
> |*Table 2:*|*Train set is RT and* |*Test set is XSTest*|
>
> ---
> ### Q3) Layer-Specific vs. Unified Alignment: Scope of L-PPO
>
> In this paper, our proposed L-PPO is designed to provide safety alignment on a per-layer basis, ensuring effective reduction in the ICET vulnerability. We agree that adapting L-PPO to improve safety alignment across all layers simultaneously is a very interesting direction and a natural next step. However, since each layer of the image encoder learns distinct representations, enforcing alignment across all layers at once could compromise the fine-grained safety guarantees that L-PPO provides, and might also affect overall utility. Therefore, we leave this for future research, where techniques such as cross-layer knowledge transfer or global safety constraints could be explored to extend the capabilities of L-PPO across the entire model.
>
> ### References
>
> [1] TRL: Transformer Reinforcement Learning, HuggingFace\
> [2] Huanjin et al. Dense Connector for MLLMs., Neurips 2024\
> [3] Skean et al. Layer by Layer: Uncovering Hidden Representations in Language Models., ArXiv 2025\
> [4] Chakraborty et al. Cross-Modal Safety Alignment: Is textual unlearning all you need?, EMNLP 2024\
> [5] Ouyang et al. Training language models to follow instructions with human feedback., 2022, arXiv\
> [6]  Röttger et al. XSTEST: A Test Suite for Identifying Exaggerated Safety Behaviours in Large Language Models, NAACL 2024\
> [7] Yu et al. Robust LLM safeguarding via refusal feature adversarial training, ICLR 2025\
> [8] Chen et al. SelfIE: Self-Interpretation of Large Language Model Embeddings., 2024, arXiv

---

> > ### Comment · Reviewer_WDjm · 2025-04-04
> >
> > Thank you for clarifying the effect of the hyperparameters, and the additional experiments on VQA and over-rejection. Your findings show that L-PPO remains effective while preserving overall utility, which addresses our primary concerns.
> >
> > I maintain my positive score and recommend the paper for acceptance.

---

> > > ### Author Response · Authors · 2025-04-04
> > >
> > > We sincerely appreciate you recommending the paper for acceptance and we’re pleased to hear that your concerns have been resolved. We are happy to address any further questions you may have.

---

### Decision · Program_Chairs · 2025-05-01

**Decision:**

Accept (spotlight poster)

**Comment:**

This paper reveals a critical vulnerability in vision-language models (VLMs): extracting intermediate-layer embeddings from the image encoder can circumvent the model’s safety alignment—even when processing safe input images. To counter this, the authors introduce Layer-Wise PPO (L-PPO), an adaptation of Clipped-Proximal Policy Optimization (Clip-PPO) that employs layer-wise multimodal reinforcement learning from human feedback (RLHF) to restore safety alignment. Extensive experiments across multiple VLMs and diverse datasets demonstrate that L-PPO significantly reduces harmful outputs.

Following the rebuttal phase, the authors have effectively addressed the initial concerns regarding additional results and novelty. The paper now receives three Weak Accept ratings and one Clear Accept. The AC agrees with the reviewers and recommends incorporating the rebuttal discussions into the final version.